# Loop-Mediated Isothermal Amplification (LAMP): Potential Point-of-Care Testing for Vulvovaginal Candidiasis

**DOI:** 10.3390/jof9121159

**Published:** 2023-12-02

**Authors:** Meng Li, Xiangyu Jin, Qingyun Jiang, Hongbo Wei, Anni Deng, Zeyin Mao, Ying Wang, Zhen Zeng, Yifan Wu, Shuai Liu, Juhyun Kim, Xiaoqian Wang, Ying Liu, Jun Liu, Wenqi Lv, Leyang Huang, Qinping Liao, Guoliang Huang, Lei Zhang

**Affiliations:** 1School of Clinical Medicine, Tsinghua University, Beijing 100084, China; li-m21@mails.tsinghua.edu.cn (M.L.);; 2Department of Obstetrics and Gynecology, Beijing Tsinghua Changgung Hospital, School of Clinical Medicine, Tsinghua University, Beijing 102218, China; 3Department of Biomedical Engineering, School of Medicine, Tsinghua University, Beijing 100084, China; xy-jin13@hotmail.com (X.J.);; 4Beijing Institute of Spacecraft System Engineering, China Academy of Space Technology, Beijing 100094, China

**Keywords:** vulvovaginal candidiasis, loop-mediated isothermal amplification, point-of-care testing, molecular diagnosis, female genital tract

## Abstract

Purpose: The aim of this study is to establish a loop-mediated isothermal amplification (LAMP) method for the rapid detection of vulvovaginal candidiasis (VVC). Methods: We developed and validated a loop-mediated isothermal amplification (LAMP) method for detecting the most common *Candida* species associated with VVC, including *C. albicans*, *N. glabratus*, *C. tropicalis*, and *C. parapsilosis*. We evaluated the specificity, sensitivity, positive predictive value (PPV), negative predictive value (NPV), and Kappa value of the LAMP method to detect different *Candida* species, using the conventional culture method and internal transcribed spacer (ITS) sequencing as gold standards and smear Gram staining and real-time Rolymerase Chain Reaction (PCR) as controls. Results: A total of 202 cases were enrolled, of which 88 were VVC-positive and 114 were negative. Among the 88 positive patients, the fungal culture and ITS sequencing results showed that 67 cases (76.14%) were associated with *C. albicans*, 13 (14.77%) with *N. glabratus*, 5 (5.68%) with *C. tropicalis*, and 3 (3.41%) with other species. Regarding the overall detection rate, the LAMP method presented sensitivity, specificity, PPV, NPV, and Kappa values of 90.91%, 100%, 100%, 93.4%, and 0.919, respectively. Moreover, the LAMP had a specificity of 100% for *C. albicans*, *N. glabratus*, and *C. tropicalis*, with a sensitivity of 94.03%, 100%, and 80%, respectively. Moreover, the microscopy evaluation had the highest sensitivity, while the real-time PCR was less specific for *C. albicans* than LAMP. In addition, CHROMagar Candida was inferior to LAMP in detecting *non-albicans Candida* (NAC) species. Conclusions: Based on the cost-effective, rapid, and inexpensive characteristics of LAMP, coupled with the high sensitivity and specificity of our VVC-associated *Candida* detection method, we provided a possibility for the point-of-care testing (POCT) of VVC, especially in developing countries and some laboratories with limited resources.

## 1. Introduction

Vulvovaginal candidiasis (VVC) is a common infection of the female genital tract, usually caused by *Candida albicans* but occasionally by other *Candida* species or yeasts [1]. Approximately 75% of women have at least one VVC episode, and 40–45% have two or more episodes in their lifetime [2]. Most VVC patients can be diagnosed based on typical symptoms, such as vulvar pruritus or burning, dyspareunia, and thick curdy vaginal discharge. However, laboratory examination is necessary to identify patients with atypical symptoms and assist in the clinical diagnosis. Currently, the routine methods for *Candida* spp. identification includes the gold standard culture method and morphological examination (KOH wet mount and smear Gram staining) [3]. In recent years, as the culture method is time-consuming and morphology does not provide species-level identification, many molecular and functional methods have emerged [4]. Several PCR (polymerase chain reaction)-based methods have been established and have demonstrated excellent performance for detecting *Candida* infections in the clinic. Khaksar Baniasadi A et al., using PCR-RFLP, and the PCR extension of the HWP1 gene, revealed that 52 yeast isolates from 119 symptomatic patient samples belonged to eight *Candida* species [5]. Shi Y et al. successfully identified over 20 *Candida* species from vaginal samples, compared to traditional methods that only detected 11 species, thus validating the accuracy of molecular diagnosis [6]. Although PCR is highly respected for its high sensitivity and short reaction time, certain nested PCR assays can occasionally yield unexplained false-positive clinical results due to their extreme sensitivity. For this reason, most PCR tests for yeasts are too sensitive to be FDA-approved [7]. Serological methods are non-DNA-based techniques that have been developed and are commercially available for the detection of circulating *Candida* antigens, such as latex agglutination, enzyme-linked immunosorbent assay, immunoblotting, dot immunoassays, liposomal immunoassays, radioimmunoassay and so on. However, serological methods lack sensitivity and specificity and also have a delayed diagnosis. Additionally, due to its susceptibility to temperature and pH, the dry chemical enzyme method has not been applied on a large clinical practice scale [8]. Furthermore, the most commonly implicated non-*albicans Candida* species (NAC) include *N. glabratus*, *C. tropicalis*, *C. krusei*, and *C. parapsilosis*, with an increasing trend over time [9]. Due to the growing azole resistance of *C. albicans* and the intrinsically resistant NAC species, the rapid and accurate identification of diverse *Candida* species in clinical microbiology and public health laboratories seems imperative [10].

Loop-mediated isothermal amplification (LAMP) is a “simple, rapid, accurate, and low-cost” method for gene amplification, which can amplify nucleic acids (usually within 1 h) under isothermal (60–65 °C) conditions [11]. Compared with conventional PCR, LAMP omits template thermal denaturation, temperature cycling, agarose gel electrophoresis, and UV light irradiation, being suitable for POCT (point-of-care testing). Another advantage of LAMP is its high specificity provided by four (or six) different primers binding to six (or eight) specific locations on the target gene [12]. Over the past decade, LAMP has been increasingly used for pathogen detection for various common clinical infectious diseases, mainly respiratory infections, foodborne diseases, and neurological infections [13,14,15]. In addition, LAMP has been used to detect diverse viruses and played an important role in detecting SARS-CoV-2 during the COVID-19 pandemic [16]. It has been well established through various studies that LAMP is highly suitable for clinical practice due to its ease of use, low cost, and resistance to inhibitors in complex samples (such as blood). However, LAMP is rarely used in diagnosing gynecological infections, and there are no relevant studies for VVC.

Given the current diagnostic urgency surrounding these pathogens, we evaluated the clinical performance of LAMP for detecting different *Candida* species. We evaluated the LAMP specificity, sensitivity, positive predictive value (PPV), negative predictive value (NPV), and Kappa values compared to the traditional culture method, smear Gram staining, and real-time PCR.

## 2. Materials and Methods

### 2.1. Study Design and Enrolment

Vaginal secretions were prospectively collected from the outpatient department of Obstetrics and Gynecology in Beijing Tsinghua Changgung Hospital between 1 and 30 June 2022. Before beginning this study, ethical clearance was granted by the Ethics Committee of the Beijing Tsinghua Changgung Hospital (19204-0-03). Women between 18 and 65 years with a subjective complaint of lower genital tract who had not received antimicrobial therapy for at least 10 d before examination were eligible for enrollment.

### 2.2. Sample Collection and Processing

Vaginal secretions were obtained from the lateral wall of the upper 1/3 of the vagina and immediately smeared on the slide for later microscopy. The remaining swabs were washed in 1 mL sterile PBS and vigorously vortexed, then the rinsing solution was taken for yeast culture (10 µL), real-time PCR (200 µL), and LAMP assay (200 µL). Specific implementation steps are shown in Figure 1.

### 2.3. Cultivation and Internal Transcribed Spacer (ITS) Sequencing

The cultivation was performed with CHROMagar Candida (Liofilchem, Roseto degli Abruzzi, Italy) at 37 °C for 20 to 48 h. The CHROMagar Candida can identify different species according to the color produced by the corresponding enzymatic hydrolysis. We selected monoclonal colonies to amplify ITS ribosomal RNA (rRNA) with the following primers: ITS1: 5′-TCCGTAGGTGAGTGCGG-3′; ITS4: 5′-TCCTCCGCT1TATTGATATGC-3′. Standard PCRs were carried out using a 2× Flash PCR MasterMix (CWBIO, Taizhou, China) following the manufacturer’s instructions. PCR experiments were performed using a T100™ Thermal Cycler (Bio-Rad Laboratories, Inc. Hercules, CA, USA) for one cycle of 3 min at 98 °C followed by 30 cycles of denaturation at 94 °C for 10 s, annealing at 58 °C for 30 s and extension at 72 °C for 30 s, with a final extension at 72 °C for 1 min. The PCR products were then purified and subjected to sequence analysis using BLAST (https://blast.ncbi.nlm.nih.gov/Blast.cgi, accessed on 28 July 2022) and Clustal Omega (https://www.ebi.ac.uk/Tools/msa/clustalo/, accessed on 28 July 2022) to identify specific *Candida* species.

### 2.4. Microscopy

Vaginal discharge smears were examined under Nikon eclipse 80i (Nikon, Tokyo, Japan) after Gram staining. Ten randomly selected high-power fields were examined. *Candida* hyphae or spores were considered positive, and their absence was considered negative.

### 2.5. Real-Time PCR

Herein, we used a commercial *C. albicans* nucleic acid detection kit (Shanghai ZJ Bio-Tech Co., Ltd., Shanghai, China; batch number P20220401) for real-time PCR assay, following the manufacturer’s instructions. Briefly, the rinsing solution was centrifuged to remove the supernatant and resuspended with 100 µL DNA extraction buffer, followed by heating at 100 °C for 10 min. Next, the supernatant containing the extracted DNA was used as the template for real-time PCR on CFX96 (Bio-Rad Laboratories, Inc., Hercules, CA, USA). Each real-time PCR was carried out in a 40 μL reaction volume containing 36 μL Master Mix (35 μL reaction mix, 0.4 μL enzyme mix, and 1 μL internal control) and 4 μL extracted DNA. Amplifications were performed as follows: 1 cycle of 2 min at 94 °C, 40 cycles of 15 s at 93 °C, and 1 min at 60 °C. At least 3 independent replicates of the real-time PCR were performed.

### 2.6. LAMP Assay

Further, we conducted the LAMP assay combined with rapid sample processing methods. The whole process could be finished within 70 min (Figure 2). The template used in the LAMP assay was crude DNA of the sample, and the sample processing method included pre-treatment and lysis steps. In the pre-treatment step, samples were centrifugated at 7000 g for 1 min, and the pellets were resuspended with 300 µL hydrolysis buffer (1 M sorbitol, 0.1 M EDTA, 0.01 M 2-Mercaptoethanol, and 100 U lyticase in 1× PBS buffer), followed by incubation at 30 °C for 20 min. In this step, *Candida* cell walls were hydrolyzed by lyticase, leading to a protoplast state. In the lysis step, pretreated samples were centrifugated at 7000 g for 1 min. Each pellet and 0.1 g glass beads (d = 100 µm) were resuspended with 300 µL TE buffer, followed by vibration at 1000 g and incubation at 99 °C for 5 min. In this step, the nucleic acids of *Candida* cells were released through thermal denaturation and bead-beating.

Finally, the supernatant was used for the LAMP assay with the WarmStart LAMP Kit (New England Biolabs, Inc., Ipswich, UK). Reaction systems were set at 10 µL, including 5 µL 2× Master Mix, 0.2 µL fluorescent dye, 1 µL LAMP primer mix, and 3.8 µL template supernatant. The LAMP primers used here are shown in Appendix A. To design species-specific primers for each target, ITS sequences of *Candida albicans* (GenBank no. NC_032096.1), *Nakaseomyces glabrata* (GenBank no. NC_006035.2), *Candida parapsilosis* (GenBank no. NW_023503284.1), and *Candida tropicalis* (GenBank no. CP047875.1) were obtained from the NCBI GenBank database (https://www.ncbi.nlm.nih.gov/genbank/, accessed on 28 July 2022). The further primer selection process includes five steps: selection of specificity, selection of sensitivity, verification of sensitivity, verification of secondary structure, and verification of specificity, which is called “Five-step screening” (Appendix A). In the LAMP assays, the concentrations of FIP/BIP and F3/B3 primers were 1.6 μM and 0.2 μM, respectively, in a total reaction volume of 10 μL. The LAMP reaction was conducted at 65 °C for 40 min using ABI7500 (Thermo Fisher, Waltham, MA, USA). Fluorescent signals were collected every minute. In the same way, we performed at least 3 independent replicates for each sample. For the specific screening process and results, refer to Jin XY et al. [17]. In our preliminary experiment, *Candida* was detected in 18 VVC clinical samples, including 9 *C. albicans*, 5 *N. glabratus*, 3 *C. tropicalis* and 1 *C. parapsilosis*, with a sensitivity and specificity of 100% [17]. At the same time, in order to eliminate the effects of contamination from sample processing step and amplification, we completed the step of sample processing, step of LAMP mixture preparation, step of template supernatant addition into the mixture, and step of real-time detection in four independent spaces. Each space was cleaned using 1% NaClO solution and radiated by UV (ultraviolet) light for 2 h every day. The amplicons were sealed in the tubes and put into Ziplock bags without any opening after the detection. In addition, uracil–DNA–glycosylase-supplemented (UDG) LAMP was used when the primers were screening at the beginning [18]. AxyPrep™ PCR Cleanup Kit (Corning Inc., New York, NY, USA) was used to eliminate the potential contamination after each step was completed. Meanwhile, negative controls were also carried out in each assay to ensure there were no false-positive results.

### 2.7. Statistical Analysis

SPSS software (version 23.0) was used for statistical analysis of the data, the count data were expressed as [case (%)], and the comparison was made by X^2^ test or Fisher’s exact probability method. Taking the conventional culture method plus ITS sequencing results as the gold standard, the diagnostic value of LAMP, smear Gram staining and real-time PCR examinations for VVC was evaluated using a four-table (Kappa test). *p* < 0.05 was considered to be statistically significant.

## 3. Results

In the present study, we enrolled 202 patients with a mean (±SD) age of 38.2 (±10.1) years. Among them, 88 were diagnosed with VVC, and 114 were negative (the yeast culture was reported positive even when only one colony had grown). Among the 88 positive patients, the fungal culture and ITS sequencing results showed that 67 cases (76.14%) were *C. albicans*, 13 (14.77%) were *N. glabratus*, 5 (5.68%) were *C. tropicalis*, and 3 (3.41%) were others species, including *Trichosporon asahii*, *Candida nivariensis*, and *Clavispora lusitaniae.* The microscopy correctly identified 81 of the 88 VVC patients (sensitivity 92.0%), superior to LAMP (sensitivity 90.9%). Moreover, the LAMP assay presented specificity, PPV, NPV, and Kappa values of 100%, 100%, 93.4%, and 0.919, respectively (Table 1).

Regarding the diagnostic accuracy for *C. albicans*, ITS sequencing was used as the gold standard based on yeast culture. The real-time PCR correctly detected 62 positive cases out of 67 *C. albicans* VVCs, with a sensitivity of 92.5%, but diagnosed 13 false positives among the 135 negative cases (specificity of 90.4%) (Table 2). Furthermore, the sensitivity and specificity of CHROMagar Candida for *C. albicans* were higher than the real-time PCR (95.5% and 99.3%, respectively) (Table 2). Notably, the LAMP assay presented a specificity of 100% for *C. albicans* detection, as well as a higher sensitivity than the real-time PCR (94.0%) (Table 2).

The sensitivity and specificity of the LAMP assay for 13 cases of *N. glabratus* were 100% (Table 3). The sensitivity of CHROMagar Candida to *N. glabratus* was worse than *C. albicans*: 10 of 13 *N. glabratus* cases were detected (76.9%), with PPV and NPV of 83.3% and 98.4%, respectively. *C. tropicalis* results are shown in Table 4. The LAMP assay identified 4 of 5 *C*. *tropicalis*-positive patients (sensitivity of 80%). Additionally, LAMP had a specificity of 100% f or *C. tropicalis*. CHROMagar Candida detected only three positive and one false positive case, with sensitivity and specificity of 60% and 99.5%, respectively. It is worth mentioning that in 88 positive patients, *C. parapsilosis* was not detected by either LAMP or culture plus ITS sequencing.

Furthermore, we evaluated the chromogenic accuracy of CHROMagar Candida using ITS sequencing. Nine cases presented inconsistent color indicator and sequencing results: 65 of 67 green strains were *C. albicans* (97.01%), and the other 2 cases were *N. glabratus* (2.98%); among the 11 purple strains, 9 were *C. albicans* (81.82%), and 2 were *C. tropicalis* (18.18%) (Appendix A). In addition, the LAMP assay did not present detection errors for *C. albicans*, *N. glabratus*, or *C. tropicalis*, except for very few missed detections (four *C. albicans*, one *C. tropicalis*, and three other species) (Appendix A).

## 4. Discussion

VVC often defined by a combination of non-specific vaginal symptoms and the presence of yeast (a common vaginal commensal), so the diagnosis of VVC is not straightforward. Moreover, approximately 50% of women with a positive NAC culture will have mild or no symptoms, and successful treatment is often complex. Hence, a definitive diagnosis is crucial [19]. Recurrent vulvovaginal candidiasis (RVVC) refers to three or more symptomatic VVCs within one year and can adversely affect quality of life, mental health, sexual activity, and even socioeconomic burden [20,21]. Typically, RVVC requires antifungal maintenance therapy with azole drugs to attenuate the recurrence of the disease [22]. The most common risk factors among VVC patients are long-term use of antibiotics, diabetes mellitus, hyperestrogenemia, sexual activity, contraceptives, pregnancy, oral contraceptives (OCPs), an intrauterine device (IUD), and so on [23,24]. Additionally, genetic variation in host genes may influence VVC susceptibility. Increased susceptibility was associated with a unique 12/9 genotype based on a genetic survey involving over 800 patients. Also, inflammatory markers in the vaginal fluid showed an increase in IL-1β and a decrease in IL-1RA in both asymptomatic and symptomatic states [25]. Therefore, the early diagnosis of *C. albicans* or non-*albicans* VVCs/RVVCs followed by prompt administration of antifungal therapy is crucial.

Surveillance on incidence rates related to VVC is limited since approximately 10% cases are asymptomatic, while data collected usually only relate to women who seek medical attention at hospitals. Several studies have reported the strain distribution of patients presenting with symptoms of VVC, and the worldwide prevalence of VVC varies across regions. *C. albicans* was the most common strain globally, with the percentage ranging from 47% to 89% [26]. Zeng et al. reported that 89% of VVC patients were infected with *C. albicans* and 11% with *N. glabratus* [27]. In addition, Ozcan et al. found that *C. albicans* had the highest percentage (57%), followed by *N. glabratus* (28%) [28]. Similar to the above studies, the probability of *C. albicans* and *N. glabratus* infections was the highest in this experiment, 76.14% and 14.77%, respectively. Prior studies have noted that the diagnosis of VVC and RVVC mostly rely on the self-report of physician diagnosis [4,29]. The symptoms of VVC—itching, inflammation, and discharge—overlap with those of other common vaginal infections, especially bacterial vaginosis (BV), and therefore symptom-based diagnosis has low sensitivity and specificity (both over- and under-diagnosis are possible) [30,31]. In clinical practice, yeast culture is the reference standard for VVC diagnosis, which is more sensitive than microscopic examination and can be used for strain identification and drug sensitivity test to guide clinical medication [32]. However, due to its time-consuming characteristics, it cannot promptly provide a treatment basis for patients. Despite demonstrating that CHROMagar Candida was highly sensitive and specific to various *Candida* species, it still requires incubation for 20–48 h. In addition, the sensitivity and specificity of CHROMagar Candida for the diagnosis of *C. albicans* were greater than 95%, but the sensitivity for *N. glabratus* and *C. tropicalis* was relatively poor. Additionally, we detected three samples as pink colonies in 88 positive patients by the CHROMagar Candida and diagnosed as *C. krusei*. However, at least three clones from each sample were selected for ITS sequencing, and two were confirmed to be *N. glabratus* and one was *C. albicans.* This is despite the fact that, for complicated VVC, especially RVVC, the pathogenic microorganism is generally NAC, which cannot be accurately detected by most clinical detection technologies at present. Although yeast culture requires a relatively long incubation period, it allows for the selection of effective antifungal agents through culture and drug sensitivity tests. Currently, routine microscopy examination is a common method for detecting VVC, with results available within 1 h [33]. Nevertheless, microscopy has limitations as it heavily relies on the expertise of technical personnel and can be influenced by laboratory conditions. Herein, the specificity and sensitivity of microscopy were greater than 90%, which might be attributed to the experience and technique of laboratory physicians. With the introduction of matrix-assisted laser desorption ionization-time of flight mass spectroscopy (MALDI-TOF MS), fungi can be rapidly identified [34]. However, the cultural dependence and the size/quality of the libraries used are evident limitations of this approach.

Alternatively, culture-independent analysis based on molecular biological methods might provide an alternative for the accurate and rapid identification of *Candida* species. Apart from commonly used real-time PCR methods based on species-specific probes or primers targeting specific fungal pathogens, other molecular biology-based methods are available such as peptide nucleic acid fluorescence in situ hybridization (PNA FISH) [35,36], restriction fragment length polymorphism (RFLP) [37], randomly amplified polymorphic DNA (RAPD) analysis [38], Multi locus sequence typing (MLST), and Microsatellite typing [39]. Here, the real-time PCR presented a high NPV for *C. albicans*, but the PPV was only 82.7%, related to a high false positive rate.

POCT is a clinical laboratory test performed at or near the point of care [40]. Compared to laboratory testing, POCT offers rapid turnaround of test results, leading to enhanced clinical and economic outcomes. Furthermore, POCT addresses the cumbersome steps of traditional laboratory tests (multiple steps of sample collection, transport, and processing) by bringing the laboratory close to the patient [41]. Over time, several variants of LAMP have emerged, including reverse transcription loop-mediated isothermal amplification (RT-LAMP), multiplex loop-mediated isothermal amplification (M-LAMP), and real-time observation of the product by modification of the original protocol to enable RNA analysis and multiplexed detection [42]. Notably, the LAMP assay had 100% specificity for *C. albicans*, *N. glabratus*, and *C. tropicalis* with yeast culture plus ITS sequencing as gold standards. Among the LAMP missed cases, four *C. albicans* and one *C. tropicalis*-related cases were due to low concentrations (less than 10 colonies/10 µL), and three related other species were due to no specific primers designed. In the LAMP assay, we simultaneously tested for *C. albicans*, *N. glabratus*, *C. tropicalis*, and *C. parapsilosis*. No *C. parapsilosis* was detected in the positive samples, which might be related to limited sample size or geographical population characteristics. Many studies have recognized LAMP’s importance as a cost-effective, rapid, sensitive, highly specific, and inexpensive POCT in clinical diagnosis, widely used to detect DNA and RNA viral pathogens [43,44]. For example, Kitajima et al. (2021) developed a reverse transcription (RT) LAMP for SARS-CoV-2 detection with high specificity, sensitivity, PPV, NPV, and concordance rate, being at least equivalent to RT-PCR in terms of utility and might be suitable as a POCT in more diverse settings [16]. Moreover, the LAMP assay also performed well in diagnosing *Chlamydia trachomatis*, with a sensitivity of 93.65% and a specificity of 99.31% [45]. Similarly, Sun et al. (2021) established a method for the diagnosis of *Bordetella pertussis* infections based on LAMP and nanoparticle-based lateral flow biosensor (LFB), with a Kappa value of 0.96 [46]. As for *Candida* spp. detection, data from Fallahi et al. suggest that the LAMP method is faster, more easily deployable, or more sensitive in diagnosing 70 different species of *Candida* compared to common diagnostic methods [47]. The LAMP method was used in another trial to detect *C. albicans* in 330 clinical samples, with 100% diagnostic accuracy compared to traditional clinical culture methods [48]. It is worth mentioning that LAMP is not only used to detect human pathogens, but also animal and plant pathogens, and can even be applied in forensics [49,50,51]. Additionally, in the LAMP approach, samples are only heated and added with the corresponding reagent in a reaction tube, representing a low-resource setting [52,53]. Each DNA amplification using conventional PCR requires electrophoretic separation of the product, which is time-consuming and involves exposure to mutagenic or carcinogenic substances. In contrast, a major advantage of LAMP is its ability to rapidly detect products using a variety of methods, which can be observed with the naked eye immediately after the reaction ends, or even while the reaction is in progress, without any additional procedures [54,55]. Among these methods, the standard methods for confirming the presence of LAMP products are turbidity measurement and visual colorimetry. Other methods include the use of hydroxynaphthol blue as a metal indicator for magnesium, a colorimetric LAMP based on Eriochrome Black T, intercalated fluorescent dyes, LAMP-magnetic bead aggregates, the coffee-ring effect, bioluminescence, or electrochemical luminescence [56,57]. Therefore, this method is more suitable to meet the requirements of VVC diagnosis in developing countries and some laboratories with limited resources. However, the LAMP tubes before and after the reaction were almost unchanged when observed by the naked eye in this experiment, and real-time changes in fluorescence signals needed to be captured with the ABI7500 (Thermo Fisher, Waltham, MA, USA). In terms of cost, not only are LAMP’s equipment requirements simple, but some LAMP kits offer lyophilized or freeze-dried reagents, saving the cost of transport and storage processes. WHO (World Health Organization) states in the diagnosis of pulmonary tuberculosis (TB) that LAMP is often a cheaper and more affordable alternative to Xpert MTB/RIF (Xpert) for molecular testing [52]. Similarly, in the diagnosis of VVC, LAMP largely saves the labor costs of traditional microscopy. It is well-known that LAMP has the major advantage of performing the whole experiment at a fixed temperature, but its main disadvantage is the complexity of primer design, which makes it difficult to achieve the specificity of the PCR assay [58]. Moreover, the limitation of LAMP method in our experiment is that it does not differentiate between *N. glabrata*, *N. nivariensis* and *N. bracarensis*; nor does it distinguish between *C. parapsilosis*, *C. orthopsilosis* and *C. metapsilosis*. Another possible limitation LAMP is that it can generate false-positive results due to the carry-over from previous experiments (due to the high sensitivity), particularly when upgrading to an automated platform [59], whereas we achieved 100% sensitivity and specificity for the detection of multiple *Candida* species in this experiment. A final possible shortcoming of LAMP is that it does not determine the senstitivity/resistance profile of the fungal species, especially with NAC species becoming more dominant and biofilms associated with increased resistance to fluconazole. Therefore, it is increasingly important that we implement antimicrobial stewardship and find broad-spectrum alternative treatment options in our clinics.

## 5. Conclusions

In this study, 202 clinical samples were used to verify the LAMP method of VVC proposed by Jin XY and Li M et al. [17]. We simultaneously tested for multiple *Candida* species and achieved 100% specificity and a comparable sensitivity. Among the methods evaluated, the LAMP assay was more than 94% sensitive to *C. albicans* and *N. glabratus* but only 80% sensitive to *C. tropicalis.* This low sensitivity might be related to the small sample size. Nevertheless, our experiments were performed in a laboratory rather than a patient setting, and the interval freezing process might have influenced the results to some extent. In summary, we developed and validated an accurate and specific LAMP assay to detect the most common *Candida* species associated with VVC. However, its specificity and sensitivity for each *Candida* species need to be further validated with more samples in a patient setting to make POCT of VVC possible.

## Figures and Tables

**Figure 1 jof-09-01159-f001:**
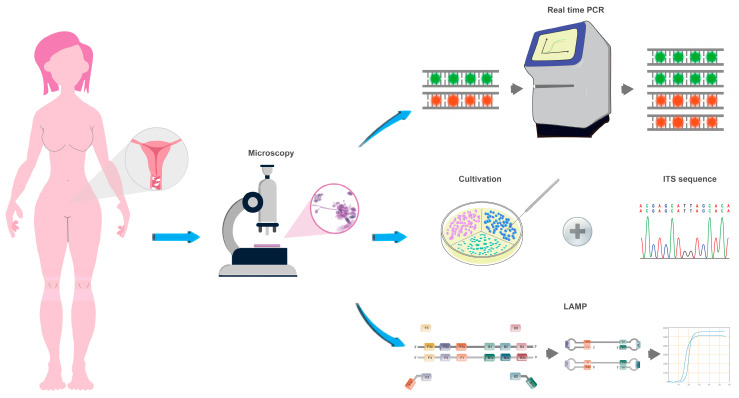
Sample processing steps.

**Figure 2 jof-09-01159-f002:**
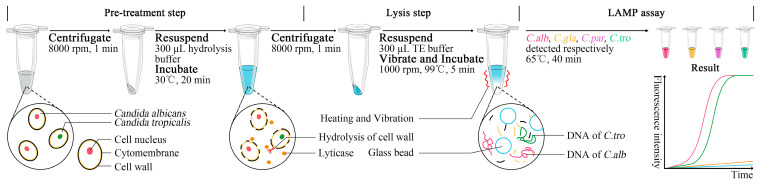
Principle of sample processing of LAMP assay for *Candida* spp. (*C. alb*, *Candida albicans*; *C. gla*, *Nakaseomyces glabrata*; *C. par*, *Candida parapsilosis*; *C. tro*, *Candida tropicalis*). In the LAMP assay section, the name of *Candida* spp., the reaction mixture in the tube, and the fluorescence intensity curve in the result with the same color represent the same detection target. For example, the green color represents the target *C. tro*.

**Table 1 jof-09-01159-t001:** Test results for *Candida* spp.

Methods	Cultivation		Sensitivity (95% CI)	Specificity (95% CI)	Positive Predictive Value (95% CI)	Negative Predictive Value (95% CI)	Kappa Value
	Positive	Negative					
Microscopy			92.05% (83.77–96.47%)	99.12% (94.50–99.95%)	98.78% (92.45–99.94%)	94.17% (94.50–99.95%)	0.919
Positive	81	1					
Negative	7	113					
LAMP			90.91% (82.38–95.71%)	100% (95.93–100%)	100% (94.29–100%)	93.44% (87.08–96.92%)	0.919
Positive	80	0					
Negative	8	114					

**Table 2 jof-09-01159-t002:** Test results for *C. albicans*.

Methods	Culture Plus ITS		Sensitivity (95% CI)	Specificity (95% CI)	Positive Predictive Value (95% CI)	Negative Predictive Value (95% CI)	Kappa Value
	Positive	Negative					
Real-time PCR			92.54% (82.74–97.22%)	90.37% (83.78–94.57%)	82.67% (71.82–90.09%)	96.06% (90.59–98.54%)	0.805
Positive	62	13					
Negative	5	122					
LAMP			94.03% (84.65–98.07%)	100% (96.55–100%)	100% (92.84–100%)	97.1% (92.34–99.07%)	0.955
Positive	63	0					
Negative	4	135					
CHROMagar			95.52% (86.62–98.84%)	99.26% (95.32–99.96%)	98.46% (90.60–99.92%)	97.81% (93.24–99.43%)	0.955
Positive	64	1					
Negative	3	134					

**Table 3 jof-09-01159-t003:** Test results for *C. glabrata*.

Methods	Culture Plus ITS		Sensitivity (95% CI)	Specificity (95% CI)	Positive Predictive Value (95% CI)	Negative Predictive Value (95% CI)	Kappa Value
	Positive	Negative					
LAMP			100% (71.66–100%)	100% (97.51–100%)	100% (71.66–100%)	100% (97.51–100%)	1
Positive	13	0					
Negative	0	189					
CHROMagar			76.92% (45.98–93.83%)	98.94% (95.83–99.82%)	83.33% (50.88–97.06%)	98.42% (95.08–99.59%)	0.787
Positive	10	2					
Negative	3	187					

**Table 4 jof-09-01159-t004:** Test results for *C. tropicalis*.

Methods	Culture Plus ITS		Sensitivity (95% CI)	Specificity (95% CI)	Positive Predictive Value (95% CI)	Negative Predictive Value (95% CI)	Kappa Value
	Positive	Negative					
LAMP			80% (29.88–98.95%)	100% (97.62–100%)	100% (39.58–100%)	99.49% (96.79–99.97%)	0.886
Positive	4	0					
Negative	1	197					
CHROMagar			60% (17.04–92.74%)	99.49% (96.77–99.97%)	75.0% (21.94–98.68%)	98.99% (96.01–99.82%)	0.659
Positive	3	1					
Negative	2	196					

## Data Availability

Data are contained within the article.

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
