# Peer review of "Loop-Mediated Isothermal Amplification (LAMP): Potential Point-of-Care Testing for Vulvovaginal Candidiasis"

_jof, 2023, doi:10.3390/jof9121159_

Round 1
Reviewer 1 Report
Comments and Suggestions for Authors
The reviewed manuscript is dedicated to the design and validation of a LAMP assay for detection of several Candida species, causative agents of vulvovaginal candidiasis. The presented results are interesting for scientists, specializing on the field of molecular diagnostics. However, a number of issues needs to be addressed before publication.
Major issues:
1. Authors are encouraged to add more information about LAMP assays for Candida detection, as many papers have been published such as 10.3390/bios13050559, 10.3389/fbioe.2022.1025083, 10.1186/s12866-022-02657-0, etc. Thus, introduction is lacking about why novel LAMP assays would be beneficial for clinical practice. Also, more information about advantages and limitations of PCR would be highly appreciated.
2. Primer’s selection is crucial for sensitivity and specificity of amplification-based tests. A separate paragraph is necessary explaining in details how LAMP primers were tested and assessed in silico. Currently, it is unclear, were these LAMP primers specific to each species or they were common. If it is multiplex LAMP, a scrupulous optimization is necessary to achieve the best possible analytical parameters.
3. Positive controls are essential parts of every diagnostic test, they allow to track performance of reagents. What were positive controls in the developed assay? Also, assessment of analytical sensitivity and specificity are absolutely necessary for NAATs.
4. Temperature gradient in LAMP could increase sensitivity of the analysis. Were such experiments conducted?
5. Examples of reaction tubes after LAMP as well as gel-electrophoresis of LAMP products would be appreciated.
6. An interesting separate question is nature of samples discrepant between microscopy and LAMP. It is legitimate to assume that LAMP-positive samples (and PCR-positive samples as well) actually contained Candida DNA, as culturing and microscopy are often less sensitive than NAATs. To test this hypothesis, amplicons from discordant samples could be sequenced.
7. Discussion lacks a comparison of the designed LAMP assay with previously reported similar LAMP tests for Candida spp.
Minor issues:
1. Minor language corrections are needed in terms of style and minor errors.
2. internal transcribed spacer (ITS) sequencing — a short description of PCR conditions is necessary.
3. All rpm values are better to be presented as g values.
4. LAMP assay — what was concentration of LAMP primers?
5. Page 8, lines 255-258: “Additionally, we detected three pink color colonies in 88 positive patients and diagnosed as C. krusei. However, at least three clones from each sample were selected for ITS sequencing, and 2 were confirmed to be C. glabrata and 1 was C. albicans.” — does it mean that from these 3 possibly C. krusei samples 2 were defined by sequencing as C. glabrata and 1 — as C. albicans?
6. Plausibly, a paragraph 272-290 could be more relevant in Introduction because it provides a lot of background information.
7. Page 8, line 287: “PNA” — PNAs are a type of artificial mimics of nucleic acids.
8. Page 9, lines 307-310: “Therefore, the 307
8.LAMP assay is a reliable alternative for accurately detecting low-copy number nucleic acids, including diagnosing epidemic diseases such as Severe Acute Respirator Syndrome (SARS) and SARS-CoV-2.” — the sentence is not linked with previous statements in the same paragraph.
9. Page 10 line 345: “This study was the first to propose a LAMP method for VVC diagnosis.” — this is arguable, as some articles reported LAMP for testing Candida spp in VVC.
Comments on the Quality of English LanguagePlease, look at the Comments and Suggestions for Authors.
Author Response
Dear reviewer,
Thank you for your letter and comments concerning our manuscript entitled “Loop-mediated isothermal amplification (LAMP): potential point-of-care testing for vulvovaginal candidiasis” (Manuscript ID: jof-2694459). Those comments are all valuable and very helpful for revising and improving our paper, as well as the important guiding significance to our researches. We have studied comments carefully and have made correction which we hope meet with approval. Revised portion are with tracked changes to highlight in the paper. The main correction in the paper and the responds to the reviewer’s comments are as following:
Major issues:
- Authors are encouraged to add more information about LAMP assays for Candida detection, as many papers have been published such as 10.3390/bios13050559,10.3389/fbioe.2022.1025083,10.1186/s12866-022-02657-0,etc.Thus, introduction is lacking about why novel LAMP assays would be beneficial for clinical practice. Also, more information about advantages and limitations of PCR would be highly appreciated.
Response: Thank you for pointing this out. We agree with this comment and have made the following changes.
- We havedescribe the relevant information of LAMP methods in the manuscript (Lines 124-129, Lines 175-176). It should be noted that the specific process of primer screening and verification have been mainly introduced in our other study (3390/bios13050559).
- We have included an introduction to the clinical benefits of the LAMP method(Lines 93-95) and the advantages and disadvantages of PCR (Lines 60-69).
- Primer’s selection is crucial for sensitivity and specificity of amplification-based tests. A separate paragraph is necessary explaining in details how LAMP primers were tested and assessed in silico. Currently, it is unclear, were these LAMP primers specific to each species or they were common. If it is multiplex LAMP, a scrupulous optimization is necessary to achieve the best possible analytical parameters.
Response: We are grateful for your constructive questioning and discussion.
- In fact, the LAMP in this study was not a multiplex LAMP, and specific LAMP primers were designed for each Candida species(supplement table 1).
- In this study, all the used LAMP primers were carefully screened. When the primers were designed, we firstly checked the performance of the primers at the theoretical level, with the Primer-BLAST of NCBI (https://www.ncbi.nlm.nih.gov/tools/primer-blast/index.cgi?LINK_LOC=BlastHome) used to verify the specificity of primers and the UNAFold (http://www.unafold.org/) used to examine the secondary structure of the primers. Afterthat, all the primers were screened through five screening steps. There were three steps designed to verify the specificity of the primers, to avoid the false positive results appeared in the reaction. In the steps of ‘selection of specificity’ and ‘verification of secondary structure’, only the primers did not produce false positive signals in negative reactions were chosen for the next-step screening. In the step of ‘verification of specificity’, only the primers did not produce false positive signals when detecting non-target samples were chosen for the clinical tests. Fortunately, as the selected primers after all the screening steps had good performance, we did not meet the problems of false-positive in this study (refer to 3390/bios13050559; Table 5).
- Positive controls are essential parts of every diagnostic test, they allow to trackperformance of reagents. What were positive controls in the developed assay? Also, assessment of analytical sensitivity and specificity are absolutely necessary for NAATs.
Response:
- About Positive controls(PC): For each test, there will be 3 tubes of sample repetition, 3 tubes of PC repetition, and 3 tubes of negative control (NC) The PC tube contains the primers that have been validated and the extracted nucleic acid (nucleic acid is 10^4 copies/μL concentration of Lactobacillus crispatus genomic nucleic acid, the design object of the primer is 16S rRNA gene of Lactobacillus crispatus).
- The sensitivity and specificity of the LAMP assayare evaluated in 3390/bios13050559, detailed in Table 4.
- Temperature gradient in LAMP could increase sensitivity of the analysis. Were such experiments conducted?
Response: Thank you for your reminding. We did not do the relevant experiments, and the reaction temperature and other conditions were in accordance with the reagent instructions.
- Examples of reaction tubes after LAMP as well as gel-electrophoresis of LAMP products would be appreciated.
Response:
- In our experiments, the LAMP tubes before and after the reaction were almost unchanged when observed by naked eyes, and it was necessary to use the ABI7500 to capture the changes in fluorescence signals in real time.
- The LAMP results are real-time fluorescence profiles, and we did not electrophorese the LAMP products.
- An interesting separate question is nature of samples discrepant between microscopy and LAMP. It is legitimate to assume that LAMP-positive samples (and PCR-positive samples as well) actually contained Candida DNA, as culturing and microscopy are often less sensitive than NAATs. To test this hypothesis, amplicons from discordant samples could be sequenced.
Response: This is a valid point (we actually did the ITS sequencing and used it as the gold standard, and most of the results were consistent with the LAMP results). We speculate that the reasons for the inconsistent results are as follows:
On the one hand, culture and microscopy are usually less sensitive than NAAT, so culture and microscopy may give false-negative results; on the other hand, the samples may contain dead bacteria.The LAMP method amplifies the target nucleic acids without distinguishing between dead and alive, but it is difficult to identify the target bacteria by culture or microscopy.
- Discussion lacks a comparison of the designed LAMP assay with previously reported similar LAMP tests for Candida spp..
Response: Your comments are very valuable. We have discuss previous LAMP tests for Candida spp. in the manuscript (Lines 333-338).
Minor issues:
- Minor language corrections are needed in terms of style and minor errors.
Response: Thank you for your comments, we have carefully checked and revised it in the manuscript.
- Internal transcribed spacer (ITS) sequencing — a short description of PCR conditions is necessary.
Response: Your comments is necessary, and we have added a short description of PCR in the methods (Lines 124-129).
- All rpm values are better to be presented as g values.
Response: We have changed all rpm to g values in the manuscript.
- LAMP assay — what was concentration of LAMP primers?
Response: Thank you for your reminding. We have list the concentration of LAMP primers in the methods (Line 171-173).
- Page 8, lines 255-258: “Additionally, we detected three pink color colonies in 88 positive patients and diagnosed as C. krusei. However, at least three clones from each sample were selected for ITS sequencing, and 2 were confirmed to be C. glabrata and 1 was C. albicans.” — does it mean that from these 3 possibly C. krusei samples 2 were defined by sequencing as C. glabrata and 1 — as C. albicans?
Response: Thank you for your comments, our expression in this paragraph is not accurate, now it has been modified in the manuscript, please refer to lines 283 to 284 for details.
- Plausibly, a paragraph 272-290 could be more relevant in Introduction because it provides a lot of background information.
Response: Your points are very reasonable and we have adjusted some of these sentence to the Introduction and made some changes (Lines 60-74).
- Page 8, line 287: “PNA” — PNAs are a type of artificial mimics of nucleic acids.
Response: We have changed “PNA (Peptide nucleic acid) FISH (Fluorescent in situ hybridization)” to “peptide nucleic acid fluorescence in situ hybridization (PNA FISH) ” (Line 303), and I don't think PNA-FISH is out of place here as an example.
- Page 9, lines 307-310: “Therefore, the LAMP assay is a reliable alternative for accurately detecting low-copy number nucleic acids, including diagnosing epidemic diseases such as Severe Acute Respirator Syndrome (SARS) and SARS-CoV-2.” — the sentence is not linked with previous statements in the same paragraph.
Response: Thank you for your reminding. This sentence is really not appropriate here, we have removed it.
- Page 10 line 345: “This study was the first to propose a LAMP method for VVC diagnosis.” — this is arguable, as some articles reported LAMP for testing Candida spp in VVC.
Response: Our statement is very inaccurate here and has been revised in the manuscript (Lines 380-381).
Reviewer 2 Report
Comments and Suggestions for Authors
Dear authors,
Li et al presented a novel way to diagnose Candida species using LAMP. The high specificity and sensitivity in combination with the fast and inexpensive characteristics support that LAMP can be a viable option as POCT.
The manuscript is nicely written and of good quality. I endorse its publication after minor revision.
Two points that can be further expanded.
- More information about the costs of the different diagnostic tools can be provided. This can further support the strength of LAMP.
- A shortcoming of the LAMP is that it does not determine the senstitivity/ resistance profile of the fungal species. Especially with NAC species becoming more dominant and biofilms associated with increased resistance to fluconazole. It is becoming more important to implement antimicrobial stewardship and find broad-spectrum alternative treatment options. In my opinion, it is important to mention this shortcoming.
Author Response
Dear reviewer,
Thank you for your letter and comments concerning our manuscript entitled “Loop-mediated isothermal amplification (LAMP): potential point-of-care testing for vulvovaginal candidiasis” (Manuscript ID: jof-2694459). Those comments are all valuable and very helpful for revising and improving our paper, as well as the important guiding significance to our researches. We have studied comments carefully and have made correction which we hope meet with approval. Revised portion are with tracked changes to highlight in the paper. The main correction in the paper and the responds to the reviewer’s comments are as following:
- More information about the costs of the different diagnostic tools can be provided. This can further support the strength of LAMP.
Response: Thank you for pointing this out. We agree with this comment and have describe the costs of different diagnostic tools (Lines 357-363).
- A shortcoming of the LAMP is that it does not determine the senstitivity/ resistance profile of the fungal species. Especially with NAC species becoming more dominant and biofilms associated with increased resistance to fluconazole. It is becoming more important to implement antimicrobial stewardship and find broad-spectrum alternative treatment options. In my opinion, it is important to mention this shortcoming.
Response: Your comments are very valuable. We have added this section to the discussion (Lines 373-377).
Reviewer 3 Report
Comments and Suggestions for Authors
Dear authors,
My general comments are:
This study concerns important information of possible non-culture methods for diagnosis of genital candidiasis in women. I agree that this common and very prevalent genital infection is mostly diagnosed without laboratory based evidence. On the other hand, emerging resistance of Candida spp. to usually proscribed antifungals additionally complicates this problem.
However, what is the most important for this study is a fact that presence of Candida spp. in genital mucosa (in interpretation of results) is not mandatory confirmation of genital candidiasis. Diagnosis of genital infection by Candida yeast required clinical observation and excluding of other possible pathogens. Molecular analyses showed that more than 64% of women have colonization of yeast of genus Candida in lower part of genital tract. So, how can this point of care test be used for diagnosis if so many positive results can be expected? With strict pathogens it is clear, but in case of Candida as commensal which can transform to pathogen it is very complicate.
In discussion (which is too long and with unnecessary facts) you have to focus on diagnosis and possible methods that can be used, you have to highlight the advantages and disadvantages of all mention methods. Be careful, microscopy cannot identify, it can detected fungal elements in material. I will give you a suggestion to present the method as possible in general diagnostics of candidosis (it will be very helpful with sterile clinical samples), not as possible useful test in diagnostics of vulvo-vaginal candidiasis, in spate you used species causative agents of this female infection.
- In Abstract you have to use abbreviations after you add it to text.
-Maybe, it is a technical error, but in some parts of article species of fungi were not written italic.
I recommend it for publication with major revision, where authors have to point out all mention facts.
It can be published as preliminary results, or pilot study.
Comments on the Quality of English LanguageModerate editing of English language required
Author Response
Dear reviewer,
Thank you for your letter and comments concerning our manuscript entitled “Loop-mediated isothermal amplification (LAMP): potential point-of-care testing for vulvovaginal candidiasis” (Manuscript ID: jof-2694459). Those comments are all valuable and very helpful for revising and improving our paper, as well as the important guiding significance to our researches. We have studied comments carefully and have made correction which we hope meet with approval. Revised portion are with tracked changes to highlight in the paper. The main correction in the paper and the responds to the reviewer’s comments are as following:
- Currently Candida glabrata is called Nakaseomyces glabrata, it would be advisable to correct the entire document.
Response: Thank you for pointing this out. We have changed all “Candida glabrata” to “Nakaseomyces glabrata”.
- The authors mention that through LAMP they can identify, among other species, N. glabrata and C. parapsilosis. But in the study the identification of N. nivariensis was achieved through sequencing of the ITS region. This result is relevant, since as the authors mention, there may be a difference in susceptibility to antifungals between different species. Therefore, I consider that the authors should mention that a limitation of the LAMP method is that it does not differentiate between N. glabrata, N. nivariensis and N. bracarensis; nor does it distinguish between C. parapsilosis, C. orthopsilosis and C. metapsilosis.
Response: We have added this section to the discussion (Lines 366-369).
- Write references in the format indicated in the instructions for authors
Response: Thank you for your reminding. We have changed the reference style.
Reviewer 4 Report
Comments and Suggestions for Authors
The manuscript “Loop-mediated isothermal amplification (LAMP): potential point-of-care testing for vulvovaginal candidiasis” presents a loop-mediated isothermal amplification (LAMP) method for detecting the most common Candida species associated with vulvovaginal candidiasis, including C. albicans, C. glabrata, C. tropicalis, and C. parapsilosis. The LAMP method presented high sensitivity, specificity, PPV, NPV, and Kappa values, which demonstrates the great diagnostic value of the method. At a methodological level, the study is well designed, the manuscript is clear, the methodology is very well described, the conclusions are based on the results presented.
However, I note there are two small details:
1. Currently Candida glabrata is called Nakaseomyces glabrata, it would be advisable to correct the entire document
2. The authors mention that through LAMP they can identify, among other species, N. glabrata and C. parapsilosis. But in the study the identification of N. nivariensis was achieved through sequencing of the ITS region. This result is relevant, since as the authors mention, there may be a difference in susceptibility to antifungals between different species. Therefore, I consider that the authors should mention that a limitation of the LAMP method is that it does not differentiate between N. glabrata, N. nivariensis and N. bracarensis; nor does it distinguish between C. parapsilosis, C. orthopsilosis and C. metapsilosis.
3. Write references in the format indicated in the instructions for authors
Author Response
Dear reviewer,
Thank you for your letter and comments concerning our manuscript entitled “Loop-mediated isothermal amplification (LAMP): potential point-of-care testing for vulvovaginal candidiasis” (Manuscript ID: jof-2694459). Those comments are all valuable and very helpful for revising and improving our paper, as well as the important guiding significance to our researches. We have studied comments carefully and have made correction which we hope meet with approval. Revised portion are with tracked changes to highlight in the paper. The main correction in the paper and the responds to the reviewer’s comments are as following:
- Results show that the LAMP test is reliable but overall inferior to microscopy (the latter failing in assessing the Candida species).In this scenario, the authors should explain and highlight (also in the introduction) the reasons why species identification is crucial for the diagnosis and for the therapeutic choices. Otherwise, the test herein proposed seems to have a very limited utility, and it might result in additional, not justified costs (several LAMP reactions per sample) compared to microscopy.
Response: Your point is very valuable. We've already described the need for Candida to be species-accurate in the introduction (Lines 78-81).
- Did the authors try to design loop primers? All the primers set proposed do not include LPs.
Response: Thank you for pointing this out. We attempted to generate loop primers using PrimerExplorer V5 (http://primerexplorer.jp/lampv5e/index.html). Despite adjusting parameters, the software failed to generate the desired loop primers. We speculate that this outcome may be attributed to the selected target gene sequence, namely the ITS sequences. Specifically, it is conceivable that the length and GC content of the sequence between F1 and F2 (or B1 and B2) may impact the design of loop primers.
- As for the sample lysis, which is performed combining an enzymatic step followed by mechanical cells disruption, I wonder if the authors tried a combination of enzymatic and chemical lysis, (I expect the latter should be very effective on protoplasts generated by the enzymatic step) or only chemical. A very rapid and effective chemical method has been recently reported.
Response: Your comments are very valuable. We acknowledge the effectiveness and rapidity of chemical lysis. However, many chemical lysis methods we investigated involve reagents containing surfactants or organic compounds such as ethanol. These substances can impact the efficiency of the LAMP reaction. Consequently, samples subjected to chemical lysis require purification before the LAMP reaction to mitigate these effects. This additional purification step introduces complexity and increases the time and operational demands. Taking these factors into consideration, we opted for a combined enzymatic and mechanical sample lysis approach.
Reviewer 5 Report
Comments and Suggestions for Authors
The manuscript reports a LAMP test for the molecular detection of Candida in vaginal smears. A separate reaction for each of the main causative agents of VVC is proposed.
Results show that the LAMP test is reliable but overall inferior to microscopy (the latter failing in assessing the Candida species).
In this scenario, the authors should explain and highlight (also in the introduction) the reasons why species identification is crucial for the diagnosis and for the therapeutic choices. Otherwise, the test herein proposed seems to have a very limited utility, and it might result in additional, not justified costs (several LAMP reactions per sample) compared to microscopy.
Minor concerns/comments:
did the authors try to design loop primers? All the primers set proposed do not include LPs.
As for the sample lysis, which is performed combining an enzymatic step followed by mechanical cells disruption, I wonder if the authors tried a combination of enzymatic and chemical lysis, (I expect the latter should be very effective on protoplasts generated by the enzymatic step) or only chemical. A very rapid and effective chemical method has been recently reported.
Comments on the Quality of English LanguagePlease revise the entire manuscript for minor errors (e.g. line 132, why "hydrolysis"? I guest authors intended lysis)
Author Response

(The authors gave the same response as above.)

Round 2
Reviewer 1 Report
Comments and Suggestions for Authors
Many thanks to authors for their efforts in editing the manuscript and replying comments. Almost all concerns have been cleared. However, before publication, some changes still seem to be necessary.
Major issues:
1. The sensitivity and specificity of the LAMP assay are evaluated in 3390/bios13050559, detailed in Table 4.
As multiplex LAMP, several simultaneous LAMP reactions were assumed in a single tube with several LAMP primer sets targeting different Candida species. Authors are encouraged to provide a brief description of the test design in the manuscript.
2. About Positive controls(PC): For each test, there will be 3 tubes of sample repetition, 3 tubes of PC repetition, and 3 tubes of negative control (NC). The PC tube contains the primers that have been validated and the extracted nucleic acid (nucleic acid is 10^4 copies/μL concentration of Lactobacillus crispatus genomic nucleic acid, the design object of the primer is 16S rRNA gene of Lactobacillus crispatus).
As positive controls, plasmids or genomic DNA were assumed that can be targeted by LAMP primers for Candida detection. Thus, LAMP performance would be assessed on independent samples.
3. The sensitivity and specificity of the LAMP assay are evaluated in 3390/bios13050559, detailed in Table 4.
Authors are encouraged to add short description of these analytical characteristics in the presented manuscript for better readability.
4. This is a valid point (we actually did the ITS sequencing and used it as the gold standard, and most of the results were consistent with the LAMP results). We speculate that the reasons for the inconsistent results are as follows.
Authors are encouraged to provide the results f ITS sequencing for discordant samples.
Author Response
Dear reviewer
We were pleased to know that our work was rated as potentially acceptable for publication in Journal, subject to adequate revision. We thank the reviewers for the time and effort that they have put into reviewing the previou sversion of the manuscript. Their suggestions have enabled us to improve our work. Based on the instructions provided in your letter, we uploaded the file of the revised manuscript. Accordingly, we have uploaded a copy of the original manuscript with all the changes highlighted by using the track changes mode in MS Word.
We would like also to thank you for allowing us to resubmit a revised copy of the manuscript.
- The sensitivity and specificity of the LAMP assay are evaluated in 3390/bios13050559, detailed in Table 4.
As multiplex LAMP, several simultaneous LAMP reactions were assumed in a single tube with several LAMP primer sets targeting different Candida species. Authors are encouraged to provide a brief description of the test design in the manuscript.
Response:
The LAMP assay employed in this study needs clarification; it is crucial to note that it is not a multiplex LAMP. Consequently, each tube contains only one set of LAMP primers. To detect four Candida species, a total of five sets of LAMP tests are conducted for each sample. Each set comprises six individual tubes of LAMP reaction targeting Lactobacillus crispatus, C. albicans, C. glabrata, C. tropicalis, and C. parapsilosis, respectively. It is noteworthy that the primer set for Lactobacillus crispatus is designated as the Positive Controls (PC) group. The following elaboration focuses on the test design using the C. albicans LAMP test group as an example.
The C. albicans LAMP test group consists of six tubes, each with a total volume of 10 µL. This includes 5 µL of 2× Master Mix, 0.2 µL of fluorescent dye, 1 µL of C. albicans-specific LAMP primer mix, and 3.8 µL of template supernatant. Among these tubes, three function as Negative Controls, wherein 3.8 µL of template supernatant is replaced with ddH2O. The remaining three tubes represent the experimental group, with 3.8 µL of template supernatant containing processed samples, as outlined in the "2. Materials and Methods" section of the manuscript.
The test design for the three groups detecting other Candida species is similar to the C. albicans LAMP test group, with the only variation being the primers used. It is crucial to emphasize that in the PC group, the 3.8 µL of template supernatant in the experimental group comprises Lactobacillus crispatus genomic DNA extracted from a pure culture using a standard reagent kit.
- About Positive controls(PC): For each test, there will be 3 tubes of sample repetition, 3 tubes of PC repetition, and 3 tubes of negative control (NC). The PC tube contains the primers that have been validated and the extracted nucleic acid (nucleic acid is 10^4 copies/μL concentration of Lactobacillus crispatus genomic nucleic acid, the design object of the primer is 16S rRNA gene of Lactobacillus crispatus).
As positive controls, plasmids or genomic DNA were assumed that can be targeted by LAMP primers for Candida detection. Thus, LAMP performance would be assessed on independent samples.
Response: Your insights are valid. The positive controls (PC) employed in this paper only serve to verify the functionality of the lamp reaction reagents. As you suggest, utilizing plasmids or genomic DNA of Candida would enable us to establish specific positive controls for each Candida strain, enhancing the rigor of our experiments. We appreciate your valuable suggestion, and we will incorporate this improvement in our future experiments.
- The sensitivity and specificity of the LAMP assay are evaluated in 3390/bios13050559, detailed in Table 4.
Authors are encouraged to add short description of these analytical characteristics in the presented manuscript for better readability.
Response: Thank you for pointing this out. We have added this section to the manuscript based on your suggestion (Lines 174-177).
- This is a valid point (we actually did the ITS sequencing and used it as the gold standard, and most of the results were consistent with the LAMP results). We speculate that the reasons for the inconsistent results are as follows.
Authors are encouraged to provide the results f ITS sequencing for discordant samples.
Response: We are very sorry, here is our expression error, all ITS sequencing results in our experiments are consistent with the LAMP results, and, we list them in Table S2 and also describe them in the manuscript Lines (241-243).
Table S2 Test results for LAMP and CHROMagar
|
Methods |
ITS sequence |
Total |
|||
|
C.albicans |
C.glabrata |
C.tropicalis |
Others |
||
|
LAMP |
|
|
|
|
|
|
C.albicans |
63 |
0 |
0 |
0 |
63 |
|
C.glabrata |
0 |
13 |
0 |
0 |
13 |
|
C.tropicalis |
0 |
0 |
4 |
0 |
4 |
|
Others |
0 |
0 |
0 |
0 |
0 |
|
CHROMagar |
|
|
|
|
|
|
C.albicans |
65 |
2 |
0 |
0 |
67 |
|
C.glabrata |
0 |
9 |
2 |
0 |
11 |
|
C.tropicalis |
0 |
0 |
3 |
1 |
4 |
|
C. krusei |
1 |
2 |
0 |
0 |
3 |
|
Others |
2 |
0 |
0 |
1 |
3 |
Reviewer 5 Report
Comments and Suggestions for Authors
The authors addressed all the reviewer's concerns, although the explanation of the reasons for which the species identification at the diagnosis moment is crucial are poorly sound. Indeed, an antifungal sensitivity test, though requiring longer times, provides more accurate information for an effective, rational (rather than empirical) therapy. I think that these aspects should be discussed in the manuscript.
Aso, the Fig.1 might be improved, using a picture of the female genitals with more realistic proportions between the different anatomical parts.
As for the lysis method, protocols much more rapid, not requiring precipitation, etc., are available.
Author Response
Dear reviewer
We were pleased to know that our work was rated as potentially acceptable for publication in Journal, subject to adequate revision. We thank the reviewers for the time and effort that they have put into reviewing the previou sversion of the manuscript. Their suggestions have enabled us to improve our work. Based on the instructions provided in your letter, we uploaded the file of the revised manuscript. Accordingly, we have uploaded a copy of the original manuscript with all the changes highlighted by using the track changes mode in MS Word.
We would like also to thank you for allowing us to resubmit a revised copy of the manuscript.
- The authors addressed all the reviewer's concerns, although the explanation of the reasons for which the species identification at the diagnosis moment is crucial are poorly sound. Indeed, an antifungal sensitivity test, though requiring longer times, provides more accurate information for an effective, rational (rather than empirical) therapy. I think that these aspects should be discussed in the manuscript.
Response: Your point is very valuable. We've already added this to the discussion in the manuscript (Lines 276-280; 289-291).
- Aso, the Fig.1 might be improved, using a picture of the female genitals with more realistic proportions between the different anatomical parts.
Response: We have made changes to Figure 1, see revised Figure 1 in the manuscript.
- As for the lysis method, protocols much more rapid, not requiring precipitation, etc., are available.
Response: I'm sorry, I don't quite understand what you mean, but are you referring to protocols that are quicker and don't need to be precipitated?
We acknowledge the effectiveness and rapidity of chemical lysis. However, many chemical lysis methods we investigated involve reagents containing surfactants or organic compounds such as ethanol. These substances can impact the efficiency of the LAMP reaction. Consequently, samples subjected to chemical lysis require purification before the LAMP reaction to mitigate these effects. This additional purification step introduces complexity and increases the time and operational demands. Taking these factors into consideration, we opted for a combined enzymatic and mechanical sample lysis approach.